# Phytochemical Screening and Characterization of Volatile Compounds from Three Medicinal Plants with Reported Anticancer Properties Using GC-MS

**DOI:** 10.3390/life14111375

**Published:** 2024-10-25

**Authors:** Emelinah Mathe, Lesibana Sethoga, Sipho Mapfumari, Oluwaseyefunmi Adeniran, Phineas Mokgotho, Jerry Shai, Stanley Gololo

**Affiliations:** 1Department of Biochemistry and Biotechnology, School of Science and Technology, Sefako Makgatho Health Sciences University, Pretoria 0208, South Africa; seyefunmisdeniran@gmail.com (O.A.); stanley.gololo@smu.ac.za (S.G.); 2Division of Pharmacology, Faculty of Pharmacy, Rhodes University, Makhanda 6139, South Africa; lesibana.sethoga@ru.ac.za; 3Department of Biomedical Sciences, Tshwane University of Technology, Pretoria 0183, South Africa; mokgothomp@tut.ac.za (P.M.); shailj@tut.ac.za (J.S.)

**Keywords:** plant extracts, phytoconstituents, biological activities, *Annona senegalensis*, *Sutherlandia frutescens*, *Withania somnifera*

## Abstract

*Annona senegalensis pers*, *Sutherlandia frutescens* (L.), and *Withania somnifera* (L.) are abundant plants and widely distributed in the Limpopo, Gauteng, Kwazulu-Natal, North West and Mpumalanga provinces in South Africa. The three plants are among those used by traditional healers and herbalists in South Africa for the treatment of a variety of diseases, including cancer. The current study aimed at the phytochemical screening and characterization of volatile compounds from the three medicinal plants using GC-MS. The methanol leaf extracts were subjected to phytochemical screening using standard chemical tests to detect the presence of different classes of compounds. Volatile compounds were detected by GC-MS analysis, and detected compounds were identified by comparing the MS spectral data with those of compounds deposited in the NIST Library (NIST08). Phytochemical screening indicated the presence of different secondary metabolites such as alkaloids, quinones, steroids, cardiac glycosides, coumarins, and terpenoids in all plants. GC-MS chromatograms allowed the detection and identification of 19 volatile compounds among the three plants with known bioactivities that are important in the management of life-threatening diseases such as cancer and diabetes. The results confirm the leaves of *Annona senegalensis*, *Sutherlandia frutescens*, and *Withania somnifera* as sources of important phytochemicals and therefore justify their use for the treatment of various diseases by traditional healers.

## 1. Introduction

Medicinal plants are reported to have been widely used for centuries to treat various diseases. This usage dates back to the beginning of human civilization before there was any record keeping. Due to the ecological distribution of plants, their usage is also indigenous to different civilizations. The sustained usage of these indigenous medicines is reported to be based on generations of indigenous knowledge propagated and/or recommendations by traditional health practitioners. These traditional health practitioners are often the knowledge holders in terms of which medicinal plants can be used to treat or manage specific illnesses and conditions. As such, a number of medicinal plants are traditionally claimed to treat various conditions such as cancer, diabetes, stress as well as many other infectious and non-infectious diseases [1,2,3]. Besides primary metabolites, medicinal plants produce secondary metabolites such as alkaloids, terpenes, glycosides, tannins, saponins, steroids, and phenols which are reported to be what is responsible for the medical properties of the plants [4,5,6,7,8]. These secondary metabolites are what the scientific community evaluates, isolates and analyzes.

The interest from the scientific community to work on identifying secondary metabolites produced by plants and evaluating their potential for treating various diseases is reportedly growing exponentially [2,9,10]. This is because new molecules from medicinal plants can be very instrumental in the development of synthesized and non-synthesized molecules that help to develop and produce modern therapeutic systems [11,12,13]. Analytical methods such as thin-layer chromatography (TLC), ultraviolet (UV), nuclear magnetic resonance spectroscopy (NMR), and gas chromatography mass spectrometry (GC-MS) are some of the important tools used in scientific research for separating, identifying, and determining the structures of phytoconstituents [11,14,15]. GC-MS is usually the method of choice when analyzing volatile molecules and compounds from medicinal plants [16,17,18,19]. This technique exploits and separates compounds based on their differences in polarity and boiling points [17]. These differences give rise to different compounds being detected at different times, which results in differences in retention times [16,18]. The retention times together with spectra are used in comparison with the National Institute of Standards and Technology (NIST) data library to predict the molecule or compounds within the mixture being analyzed [16,17,18,19]. A number of studies have been carried out using GC-MS to evaluate and predict compounds responsible for the biological activities of different indigenous South African medicinal plants, where about 80% of the South African black population is reported to use medicinal-plant-based traditional treatments for medical needs at a certain stage of their lives [2,3,20,21].

The most common medicinal plants sold and utilized in South Africa include but are not limited to *Hypoxis hemerocallidea*, *Sutherlandia frutescens*, *Warburgia salutaris*, *Annona senegalensis*, and *Withania somnifera* [22,23,24,25]. The plant *A. senegalensis*, which belongs to the *Annonaceae* family, is widely distributed in the Limpopo, Mpumalanga and Kwazulu-Natal provinces in South Africa [25,26]. Reports suggest that its leaves are frequently used as a vegetable and traditionally for the treatment of various diseases such as pneumonia, insect bites, and skin cancer, while bark is used to treat wounds and snake bites [25,26,27]. It is further reported to exhibit antitumor, anti-inflammatory, as well as antioxidant activities [25,26,27,28,29]. These biological activities of A. senegalensis could be associated with the plant’s possession of essential oils, acetogenins, a variety of alkaloids, terpenoids and flavonoids [26,28,30]. The plant *S. frutescens* is a small-leafed and red-flowered shrub belonging to the *fabaceae* family and it is well distributed across the Northern, Western and Eastern Capes as well as the Kwazulu-Natal and Mpumalanga provinces of South Africa [31,32]. Various reports suggest that it contains alkaloids, glycosides, saponins, flavonoids as well as terpenoids [24,31,33,34]. It is traditionally used as a natural remedy for various health conditions, mainly fevers, coughs, colds, tuberculosis, peptic ulcers, dysentery, and diabetes [24,35]. Furthermore, it is believed to reduce inflammation and improve the immune system as well as appetite and is reported to have anticancer properties [36,37]. *W. somnifera*, belonging to the *Solanaceae* family, is a plant native to India and Pakistan but has also naturalized in most of the nine provinces of South Africa except the Northern and Western Capes [3,38]. It is traditionally used to increase both fertility and libido [27]. It is documented to have anti-inflammatory, antioxidant, hemopoietic, antitumor, and antimetastatic properties [27,38,39]. The reported phytochemistry of this plant is very broad, including steroids, alkaloids, saponins, flavonoids, tannins and phenolic compounds [40,41,42]. Traditional healers and herbalists in the republic of South Africa use *Annona senegalensis*, *Sutherlandia frutescense*, and *Withania somnifera* for the treatment of cancer. However, there is insufficient information on the volatile compounds possessed by the leaves of these plants located in South Africa. As such, the current study reports on the phytochemical screening and characterization of the volatile compounds present in the leaves of these three medicinal plants.

## 2. Materials and Methods

### 2.1. Materials

Methanol, sulphuric acid, ferric chloride, glacial acetic acid, chloroform, Molisch’s reagent, ammonia solution, sodium hydroxide, dimethyl-polysiloxane, and N,O-bis(Trimethylsilyl)trifluoroacetamide with 10% trimethylchlorosilane. For consistency all chemical used for the SHIMADZU instrument were purchased from the South African division of Merck, Modderfontein, South Africa. The other solvents were purchased from Rochelle Chemical & Lab Equipment cc, Johanessburg, South Africa.

### 2.2. The Collection and Preparation of the Plant Material

The leaves of *Annona senegalensis* (urn:lsid:ipni.org:names:72309-1), *Sutherlandia frutescens* (urn:lsid:ipni.org:names:157207-3), and *Withania somnifera* (urn:lsid:ipni.org:names:821709-1) were collected from the Botanical Gardens, Gauteng Province, South Africa. These leaves were authenticated by an indigenous knowledge system (IKS) practitioner in the botanical garden. The freshly collected leaves were thoroughly washed under running tap water and rinsed in distilled water, air-dried at room temperature, and ground to powder using a grinder (Mellerware). The powders were consequently stored in the dark until further usage. The dry leaf powder samples (50 g/each) were extracted with 500 mL of methanol in an orbital shaker overnight. The resulting extracts were filtered using Whatman No. 1 (0.45 µm) filter paper into pre-weighed beakers and evaporated at room temperature. The total mass of each extract obtained was weighed, and the percentage yield was calculated.

### 2.3. Phytochemical Screening of the Methanol Leaf Extracts

The screening for the phytochemical composition of methanol leaf extracts was carried out using the standard chemical test methods as reported by [43] with some slight modifications. The exact steps followed are as follows:(i)Testing for reducing sugars

The presence of reducing sugars was evaluated using 2 mL (1 g dissolved in 10 mL of solvent) of extract which was treated with 1 mL of Molisch’s reagent and a few drops of concentrated sulphuric acid. The formation of purple or reddish color was taken as a positive indication of the presence of reducing sugars.
(ii)Testing for tannins

This was evaluated by adding 2 mL of 5% ferric chloride to 1 mL of the methanolic extract. The mixture was left standing for 2 min. The formation of dark blue or greenish black indicated the presence of tannins.
(iii)Testing for saponins

A foam test was used, where 2 mL of extract and 2 mL of distilled water were added together. The mixture was stirred for 15 min and the formation of a stable 1 cm layer of foam indicated the presence of saponins.
(iv)Testing for glycosides

The formation of a pink color indicated the presence of glycosides when 2 mL of extract, 3 mL of chloroform and about 10% ammonia solution were added and mixed.
(v)Testing for quinones

In evaluating the presence of quinones, 1 mL of concentrated sulphuric acid was carefully added to 1 mL of the methanolic extract. The formation of a red color indicated the presence of quinones.
(vi)Testing for phenols

To evaluate the presence of phenols, 2 mL of distilled water followed by a few drops of 10% ferric chloride were added to 1 mL of the extract. Formation of a blue or green color indicated the presence of phenols.
(vii)Testing for terpenoids

Terpenoids were evaluated when 0.5 mL of the extract was treated with 2 mL of chloroform and 3 mL of concentrated sulphuric acid. Formation of a red-brown color at the interface indicates the presence of terpenoids.
(viii)Testing for cardiac glycosides

To 0.5 mL of the extract, 2 mL of glacial acetic acid and a few drops of 0.1% ferric chloride were added. This was under-layered with 1 mL of concentrated sulphuric acid. The formation of a brown ring at the interface indicated the presence of cardiac glycosides.
(ix)Testing for coumarins

To evaluate the presence of coumarins, 1 mL of 10% sodium hydroxide was added to 1 mL of the extract. The formation of a yellow color was taken as an indication of the presence of coumarins.
(x)Testing for anthraquinones

A few drops of 10% ammonia solution were added to 1 mL of the extracts to screen for the presence of anthraquinones. Guided by previous studies, the appearance of a pink precipitate was taken as a positive indication of the presence of anthraquinones
(xi)Testing for steroids

To 1 mL of extract, an equal volume of chloroform and a few drops of concentrated sulphuric acid were carefully added. The appearance of a brown ring indicated the presence of steroids.
(xii)Testing for anthracyanines

To 1 mL of the extract, 1 mL of 2 M sodium hydroxide was added and the mixture was heated for 5 min at 100 °C. The resultant formation of a bluish-green color indicated the presence of anthracyanines.

### 2.4. Gas Chromatography-Mass Spectrometry (GC-MS) Analysis

GC-MS analysis of methanol leaf extracts was carried out on a SHIMADZU QP2010 SE instrument (Shimadzu, Kyoto, Japan) with an inert cap 5MS/SIL and silica capillary column (30 mm × 0.25 ID × 1 µmdf, composed of 100% dimethyl-polysiloxane). The dried methanol extract (1 mg) was mixed with 0.1 mL of BSTFA-TMCS solution. The mixture was left in a closed cap for a duration of 24 h in an incubator with a temperature setting of 60 °C [44,45]. This mixture was then injected into the column. An electron ionization system with an ionizing energy of 70 eV was used for detection. Helium gas was used as the carrier gas at a constant flow rate of 1ml/min with an injection volume of 2 µL, injection temperature of 260 °C and ion source temperature of 230 °C. The oven temperature was set from 50 °C (isothermal for 1 min), with an increase of 20 °C/min to 180 °C (isothermal in 5 min), which then increased to 240 °C/min, ending with an increase of 20 °C/min to 180 °C (isothermal in 5 min). Mass spectra were taken at 70 eV, scanned interval of 0.3 sec and fragments from 50 to 700 *m*/*z*. The software adopted to handle mass spectra and chromatograms was GC-MS Solution version 2.6.

### 2.5. Identification of the Detected Volatile Compounds

Computer searches using the National Institute of Standards and Technology (NIST) data library were used to compare the spectrum obtained through GC-MS with the compounds present in the crude plant extracts. The mass spectrum interpretation of GC-MS was performed using NIST. The mass spectrum of unknown components was compared with that of known components stored in the NIST library and only those with an 80% or above similarity were considered as positively present. The names of the components, molecular weight and the structure of the testing material were confirmed.

## 3. Results

### 3.1. Phytochemical Screening

The results of the phytochemical screening of the methanol leaf extracts of the three medicinal plants revealed the presence of secondary metabolites with well-known health-beneficial properties in animals and humans and are shown in Table 1 below.

As can be observed from Table 1 above, all three plants evaluated in this study have returned positive indications during analysis for the presence of carbohydrates, tannins, quinones, phenols, terpenoids, cardiac glycosides, steroids and alkaloids. The results of this study *Annona senegalensis* and *Withania somnifera* methanol extracts returned positive results for coumarins. On the other hand, only *Sutherlandia frutescens* returned positive results for saponins. Anthraquinones were only found to be present on *Withania somnifera* extract.

### 3.2. Characterization of Volatile Compounds Within the Methanol Crude Leaf Extracts

GC-MS analysis was used for the detection and identification of different metabolites within the *Annona senegalesis*, *Sutherlandia frutescence* and *Withania somnifera* leaf extracts. The results showed in total the identification of 19 volatile compounds within the plants’ methanol leaf extracts. The properties of the detected volatile compounds, including the retention time, peak area (%), molecular weight, molecular formula and the nature of the compound, are shown in Table 2.

In addition, the chemical structures of the dominant volatile compounds within the methanol leaf extracts of the three medicinal plants *A. senegalensis*, *S. frutescens* and *W. somnifera* are shown in Figure 1. Furthermore, the relative distributions of the groups of volatile compounds detected within the three medicinal plants’ methanol leaf extracts are presented in Figure 2. Ultimately, the biological activities of some of the detected volatile compounds are shown in Table 3.

This study identified 19 volatile phytochemical compounds across the three plants. The detected phytoconstituents comprised alcohols, and other predominant included fatty acids, carbohydrates, aldehydes, esters and hydrocarbons as can be seen from Table 2 above. Some of the phytochemical compounds were found to be common across the different plants. Those are glycidol oxtranemethanol (alcohol), octadecanal stearaldehyde (aldehyde), decanoic acid-3-methyl (fatty acid) and n-hexadecanoic acid (fatty acid). The chemical structures of these volatile compounds within the methanol leaf extracts of the three medicinal plants *A. senegalensis*, *S. frutescens* and *W. somnifera* are shown in Figure 1, below.

The figure below summarises the nature of phytochemical compounds identified from each plant. A few of the major components found in high percentage peak were dimethylsulfone (14.57%), glycidol oxtranemethanol (10.08%), dimethyl sulfone-methane (9.33%), 1,4-cyclohexanedimethanol (5.47%) and 2-butenedionic acid-dimethyl ester (5.11%).

The results of this study show that there is an equal distribution of carbohydrates and alcohol in the *A. senegalensis*. The GC-MS appears to not have detected esters and hydrocarbons from this plant. On the second plant, *S. frutescens*, the relative distribution shows an equal distribution of carbohydrates and fatty acids while the other compound types were also detected. Lastly, with regard to *W. somnifera*, there is an equal distribution of carbohydrates and fatty acids while esters and hydrocarbons were not detected. A few major components found in high percentage peaks across the three plants were dimethylsulfone (14.57%), glycidol oxtranemethanol (10.08%), dimethyl sulfone-methane (9.33%), 1,4-cyclohexanedimethanol (5.47%) and 2-butenedionic acid-dimethyl ester (5.11%). A literature search for the biological activities associated with the GC-MS-identified compounds revealed that the three plants may be significant prospective agents for the treatment of various ailments over and above what they are used for traditionally. Table 3 below summarizes the biological activities of all the compounds which were identified using GC-MS.

## 4. Discussion

The phytochemical screening results of this study corroborate the findings of previous studies conducted on samples from different locations, where the leaves of *A. senegalensis* were reported to contain saponins, cardiac glycosides, steroids, alkaloids, flavonoids, reducing sugar, polyphenols, terpenoids and tannins [28,29]. Similarly, the alcoholic extracts from the root powder and whole plant of *W. somnifera* were also reported to contain saponins, flavonoids, starch and carbohydrates, phenols, alkaloids, saponins, glycosides, carbohydrates and terpenoids [41,50]. The leaves of *S. frutescens* are reported to contain pinitol, flavonol, flavonoid glycosides, and triterpenoid saponins [24,51]. These are the compounds that are well known to be responsible for pharmacologically significant properties such as antiallergic, anti-inflammatory, antibacterial, antiviral, antioxidant, anti-aging and anticancer activities, among others [43]. As such, plants containing these types of phytochemicals are great potential agents that could be explored in the treatment and management of communicable and non-communicable diseases the world is battling at the moment.

Studies suggest that alkaloids, due to their structures, are some of the phytochemicals that have anticancer or antitumor properties. As such, the results of this study suggest that all three plants should potentially be great anticancer agents as they all have alkaloids as part of their phytoconstituents. Some metabolites/phytochemicals that similarly have been determined scientifically and shown to have anticancer or antitumor properties are tannins, terpenoids and coumarins [52,53,54,55]. All three plants have also been positively confirmed in this study to contain these important phytochemicals, with only *S. frutescence* returning a negative outcome for coumarins as can be observed from Table 1 above. The other phytochemicals found to be present within these three plants such as cardiac glycosides and saponins are known to lower blood pressure and act as antioxidant as well as anti-inflammatory agents [7,8,56,57].

The GC-MS analysis revealed medically significant compounds among the plants evaluated in this study. Among the detected compounds, fatty acids are known for their antimicrobial and anticancer properties, while alcohols and hydrocarbons are reported to possess antibacterial, antioxidant, and antimicrobial activity [47]. Most of the identified volatile compounds are reported to possess health-beneficial bioactivities. For example, n-hexadenoic acid is an organic acid reported to have anticancer activity and can also improve energy metabolism [49,50]. The compound 2,3,5-trimethylanizole (phenol) is reported to play a role as an antioxidant agent [49]. The detection of volatile compounds within selected medicinal plants was reported in previous studies [11,34,50,58] and the results obtained in the current study are mostly in agreement with the findings reported in previous studies, although with some observed disparities. Compared to the findings from previous studies, the detection of compounds such as cyclohexanol 3,5-dimethyl, glycidol oxtranemethanol, and D-erythro-pentose-2-deoxy-D-(ribose) is reported for the first time in this study. Therefore, the results of the current study confirm the presence of volatile compounds within the extracts of the selected medicinal plant. The findings of this study are, as such, limited since only mainly volatile compounds could be detected by the technique and polar compounds could be evaluated with the use of other techniques such as LC-MS and HPLC.

## 5. Conclusions

The need to establish the phytochemical profile of medicinal plants can never be overemphasized, as it provides scientific support for the traditional credence of the plants. Phytochemical screening in this study revealed the presence of phytochemicals with health-beneficial properties within the leaves of *A. senegalensis*, *S. frutescens* and *W. somnifera*. The identified phytochemicals included reducing sugars, glycosides, alkaloids, steroids, tannins, cardiac glycosides, coumarins, terpenoids and quinones. These are significant phytochemicals to be possessed by medicinal plants given their association with abilities to alleviate numerous communicable as well as non-communicable diseases. GC-MS enabled the successful characterization of volatile compounds within the methanol leaf extracts of *A.senegalensis*, *S. frutescens* and *W. somnifera*. Some of the detected volatile compounds were alcohols, fatty acids, carbohydrates, aldehydes, esters and hydrocarbons. These results, therefore, confirm the leaves of *Annona senegalensis*, *Sutherlandia frutescens* and *Withania somnifera* as sources of pharmacologically significant phytochemicals and as such underpins their usage for treatments of diseases in traditional medicine.

## Figures and Tables

**Figure 1 life-14-01375-f001:**
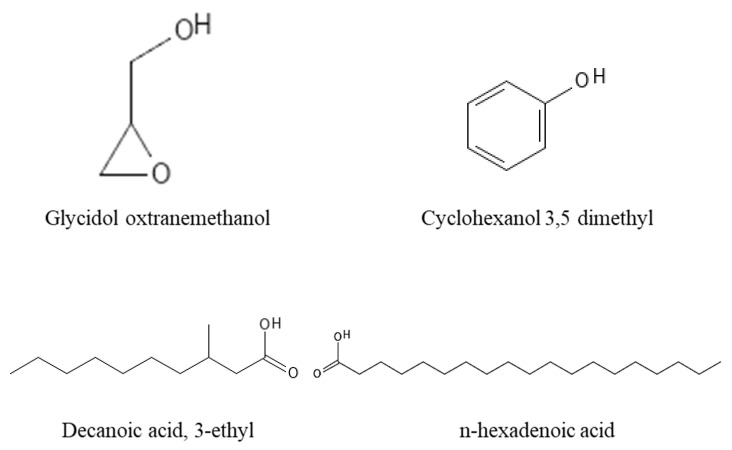
The chemical structures of the most dominant volatile compounds within the plants’ methanol leaf extracts.

**Figure 2 life-14-01375-f002:**
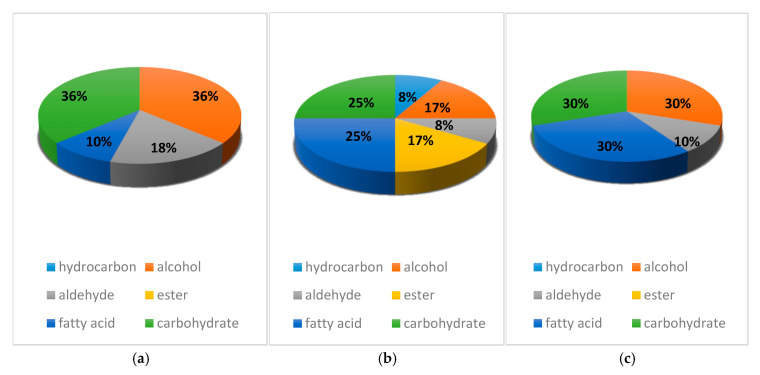
Relative distribution (%) of volatile compounds in (**a**) *A. senegalensis*, (**b**) *S. frutescents*, and (**c**) *W. somnifera* methanol leaf extracts.

**Table 1 life-14-01375-t001:** Phytochemical screening of the methanol leaf extracts of *Annona senegalensis*, *Sutherlandia frutescence* and *Withania somnifera*.

PLANT EXTRACTS	CARBOHYDRATES	TANNINS	SAPONINS	GLYCOSIDES	QUINONES	PHENOLS	TERPERNOIDS	CARDIAC GLYCOSIDES	COUMARINS	ANTHRAQUINONES	STEROIDS	ALKALOIDS
*Annona senegalensis*	+	+	-	-	+	+	+	+	+	-	+	+
*Sutherlandia frutescens*	+	+	+	-	+	+	+	+	-	-	+	+
*Withania somnifera*	+	+	-	-	+	+	+	+	+	+	+	+

(+: presence; -: absence).

**Table 2 life-14-01375-t002:** The chemical properties and relative abundance of volatile compounds detected within the crude methanol leaf extracts of *A. senegalesis*, *S. frutescens* and *W. somnifera*.

Name of Compound	Molecular Formula	MW a	Peak Area%	R Ta	Selected Plants			Nature of Compound
					AS	WS	SF	
Glycidol oxtranemethanol	C_3_H_6_O_2_	74	10.08	3.28	√	√	√	Alcohol
DimethylSulfoxide-(methane)	C_2_H_6_OS_3_	78	9.33	3.42	√			Aldehyde
Oxtrane, (butoxymethyl)-propane	C_7_H_14_O_2_	130	5.82	3.45			√	Ester
Propanoic acid, 2-hydroxyethyl ester	C_5_H_10_O_3_	118	7.71	3.58	√	√		Fatty acid
Dimethylsulfone-methane sulfonylbis	C_2_H_6_O_2_S	94	14.57	3.74	√	√		Aldehyde
2-butenedioic acid-dimethyl ester	C_6_H_8_O_4_	144	5.11	4.15			√	Ester
DL-arabinose-pentopyranose	C_5_H_10_O_5_	150	2.66	4.65			√	Carbohydrate
2,3,5-trimethylanizole (phenol)	C_10_H_14_O	150	2.36	6.61			√	Alcohol
1-hydroxy-2-pentanone	C_5_H_10_O_2_	102	4.06	7.33	√			Carbohydrate
Cyclohexanol, 3,5 dimethyl	C_8_H_16_O	128	3.93	11.92	√			Alcohol
Octadecanal stearaldehyde	C_18_H_36_O	263	1.99	11.94	√	√	√	Aldehyde
decanoic acid,3-methyl	C_11_H_22_O_2_	186	4.68	12.85	√	√	√	Fatty acid
D-erythro-pentose 2-deoxy-D-(ribose)	C_5_H_10_O_4_	134	2.31	13.62			√	Carbohydrate
n-hexadecanoic acid	C_16_H_32_O_2_	256	3.63	14.49	√	√	√	Fatty acid
Fructose	C_6_H_12_O_6_	180	4.19	15.51			√	Carbohydrate
1,4-cyclohexanedimethanol	C_8_H_16_O_2_	144	5.47	15.57	√	√		Alcohol
9,12,15-octadecatrienoic acid	C_18_H_30_O_2_	278	3.58	16.12		√	√	Fatty acid
1-pentane-2methyl	C_16_H_32_	224	1.43	21.07			√	Hydrocarbon
Phytol	C_20_H_42_O_2_	296	1.75	29.07	√		√	Alcohol

**Table 3 life-14-01375-t003:** Biological activities of some of the volatile compounds within the methanol leaf extracts of *A. senegalensis*, S. *frutescents*, and *W. somnifera* through GC-MS analysis.

Name of Compound	Biological Activity
DimethylSulfone	Anti-inflammatory and antioxidant activity [18,46]
Oxtrane (butoxymethyl)-propane	Antibacterial [47]
Propanoic acid, 2-hydroxyethyl ester	Antimicrobial [48]
2-butenedioic acid-dimethyl ester	Antimcrobial [49]
2,3,5-trimethylanizole (phenol)	Antibacterial, antioxidant [49]
1-hydroxy-2-pentanone	Anticancer, antimicrobial [47]
Octadecanal stearaldehyde	Antimicrobial [47]
Decanoic acid,3-methyl	Antibacterial [50]
n-hexadecanoic acid	Anti-flammatory, antioxidant, antidiabetic, antibacterial, anticancer, antimicrobial, and antifungal [48,49,50]
1,4-cyclohexanedimethanol	Alpha-glucosidase inhibitors [50]
9,12,15-octadecatrienoic acid	Anticancer, anti-obese, antioxidant [48]
Phytol	Antibacterial, antifungal, anticancer, antidiabetic [48,50]

## Data Availability

Data reported and covered in this study is available on request via email to: emelinah.mathe@smu.ac.za. Some GC-MS chromatograms and mass sprectra has been provided as Appendix A.

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
