# Peer review of "Phytochemical Screening and Characterization of Volatile Compounds from Three Medicinal Plants with Reported Anticancer Properties Using GC-MS"

_life, 2024, doi:10.3390/life14111375_

Round 1
Reviewer 1 Report
Comments and Suggestions for Authors
1. Please italicize all scientific names throughout the manuscript (highlighted in the manuscript).
2. For the purpose of research and publication, it is highly essential to properly identify the plant materials by submitting the voucher specimen in the Herbarium of their respective countries. Without proper identification, studies made on that plant species are not considered genuine. In this regard, the authors should provide the full accepted name of the plants under study. For this purpose, they can visit
https://www.ipni.org/ or
https://powo.science.kew.org/
3. Additionally, authors should include the full accepted name of the studied plants at least once in both the Abstract and Experimental sections.
4. While the manuscript mentions GC-MS analysis, it has failed to include the gas chromatograms for the analyzed samples. Please incorporate these chromatograms either into the main research article or as supplementary information. Additionally, the authors should provide the mass spectra of the identified major compounds from the plant extract as this would significantly enhance the data's credibility and reproducibility.
5. Under the Materials and Methods section, please include a subsection titled "Reagents." Within this subsection, list all the chemicals used in the study.
6. In Figure 1, please redraw the structures of the organic compounds to ensure consistency and uniformity.
7. In subsection 2.1, starting at line 160, the authors have mentioned that the plant extracts were derivatized. To enhance clarity the authors should create a separate heading for the derivatization process. This dedicated section should include a detailed description of all relevant steps, including reagents, reaction conditions, and any purification procedures. Also, include the reference for the derivatization process.
8. Line 46 to 61 represents general information so please rewrite to be specific.
9. The Introduction section should be revised to provide a comprehensive overview of the existing literature on the studied plants. Additionally, the authors should clearly articulate the novelty of their work, highlighting the unique contributions of their work in this particular field.
10. Avoid repeating words from the title in the keywords. Additionally, refrain from using abbreviations in the keyword list.
11. The Discussion section should be expanded to provide a more in-depth explanation of the findings. This includes elaborating on the implications of the results, comparing them to previous studies, and addressing any limitations. This will enhance the overall understanding and significance of the research.
12. Given the authors' claims of anticancer properties in the plant materials and identified compounds, it is essential to include the results of any biological studies conducted to evaluate these properties. This will provide concrete evidence supporting the potential therapeutic applications of the compounds.
13. The title contains the term "anti-cancer," which is misleading as the work does not present any direct evidence of anticancer activity. It is recommended to revise the title to accurately reflect the scope and findings of the study.
14. Please arrange the reference section as per the journal’s guidelines.
15. Please address the comments provided in the manuscript as well.

Reviewer 2 Report
Comments and Suggestions for Authors
The manuscript presents phytochemical screening and characterization of some compounds from three medicinal plants with anticancer properties. In my opinion it could present a great interest for the readers of this journal and could be published after a major revision.
As comments/sugestions:
1. In order to meet the standard of this journal, the qualitative characterization of the active principles identified in these plants should be completed by other studies (eg HPLC) that also ensure their quantitative analysis.
2. It is also necessary to identify the compounds that can be isolated and their physico-chemical characterization, especially regarding their solubility and their stability in biological environments.
Round 2
Reviewer 1 Report
Comments and Suggestions for Authors
Please add the reference for the derivatization process.
Author Response
Comment 1: Please add the reference for the derivatization process.
Response 1: Thank you for bringing this to our attention. We have added the two method references in the body of the main document, as can be seen from line 213 on page 6. The references become references [44] and [45].
Reviewer 2 Report
Comments and Suggestions for Authors
The authors completed the manuscript respect with the reviewer suggestions.
In my opinion the manuscript can be published in present form
Author Response
Comments :
The authors completed the manuscript respect with the reviewer suggestions.
In my opinion the manuscript can be published in present form
Response: Thank you